# Downregulation of γ-Catenin by miR-195-5p Inhibits Colon Cancer Progression, Regulating Desmosome Function

**DOI:** 10.3390/ijms25010494

**Published:** 2023-12-29

**Authors:** Emanuele Piccinno, Viviana Scalavino, Nicoletta Labarile, Giusy Bianco, Maria Teresa Savino, Raffaele Armentano, Gianluigi Giannelli, Grazia Serino

**Affiliations:** National Institute of Gastroenterology S. De Bellis, IRCCS Research Hospital, Via Turi 27, 70013 Castellana Grotte, BA, Italy; emanuele.piccinno@irccsdebellis.it (E.P.); viviana.scalavino@irccsdebellis.it (V.S.); nicoletta.labarile@irccsdebellis.it (N.L.); giusy.bianco@irccsdebellis.it (G.B.); maria.savino@irccsdebellis.it (M.T.S.); raffaele.armentano@irccsdebellis.it (R.A.); gianluigi.giannelli@irccsdebellis.it (G.G.)

**Keywords:** miR-195-5p, CRC progression, γ-catenin, JUP, proliferation, invasion

## Abstract

Desmosomes are essential structures for ensuring tissue functions, and their deregulation is involved in the development of colorectal cancer (CRC). JUP (γ-catenin) is a desmosome adhesion component that also acts as a signaling hub, suggesting its potential involvement in CRC progression. In this context, we recently demonstrated that miR-195-5p regulated JUP and desmosome cadherins expression. In addition, miR-195-5p gain of function indirectly modulated the expression of key effectors of the Wnt pathway involved in JUP-dependent signaling. Here, our purpose was to demonstrate the aberrant expression of miR-195-5p and JUP in CRC patients and to functionally characterize the role of miR-195-5p in the regulation of desmosome function. First, we showed that miR-195-5p was downregulated in CRC tumors compared to adjacent normal tissue. Then, we demonstrated that JUP expression was significantly increased in CRC tissues compared to adjacent normal tissues. The effects of miR-195-5p on CRC progression were assessed using in vitro transient transfection experiments and in vivo miRNA administration. Increased miR-195-5p in colonic epithelial cells strongly inhibits cell proliferation, viability, and invasion via JUP. In vivo gain of function of miR-195-5p reduced the numbers and sizes of tumors and significantly ameliorated the histopathological changes typical of CRC. In conclusion, our findings indicate a potential pharmacological target based on miR-195-5p replacement as a new therapeutic approach in CRC.

## 1. Introduction

Colorectal cancer (CRC) is one of the most lethal and prevalent malignancies in the world. Surgery and chemotherapy have been the only option for cancer patients, but unfortunately the prognosis of CRC has remained poor, especially for patients with metastatic lesions. Targeted therapy is a new approach that has increased the overall survival of CRC patients [1]. A better understanding of the molecular mechanism of colorectal cancer progression may lead to new specific therapeutic strategies.

Desmosomes together with tight and adherence junctions form junctional complexes, an intercellular structure that guarantees intestinal homeostasis [2]. The adhesive junctions contribute to maintain the physical integrity of the mature epithelium and also respond to mechanical and biochemical signals by regulating cellular movement, intracellular signaling, and gene transcription [3,4,5].

Dysregulation of the desmosome complex plays a critical role in malignant transformation and metastasis [6,7]. Desmosome components can act as both a tumor promoter and a tumor suppressor, suggesting that desmosomal genes have cell type-specific functions that might not always be related to altered cell adhesion but could also trigger abnormal cell signaling [8,9]. The loss of desmosome proteins contribute to cell proliferation by enhancing Akt/β-catenin signaling and facilitate malignant transformation, providing clear evidence of tumor growth in immunodeficient mice [10]. Alternatively, another study showed that loss of the intestinal desmosomal cadherin resulted in decreased epithelial cell proliferation and suppressed xenograft tumor growth in mice, interfering downstream with the epidermal growth factor receptor (EGFR) transduction [11]. In colorectal cancers, loss of cell junctions has been characterized as a unique trait of cells undergoing the epithelial-mesenchymal-transition (EMT) and is usually concomitant with deregulation of the Wnt signaling pathway that is aberrantly activated [12]. 

JUP (Junction plakoglobin or γ-catenin) is an essential desmosome component that, in addition to its direct role as a physical linker of the actin cytoskeleton to desmosomal cadherins (desmoglein and desmocollin), plays a central role in signal transduction and the regulation of gene expression [13].

As a member of the armadillo family protein, γ-catenin is a close homolog of ß-catenin, that shares several common protein partners with ß-catenin and executes some of the same functions. Recent evidence reported the transcriptional activity of JUP in the Wnt signaling cascade, suggesting that JUP can function as an oncogene when its expression is deregulated in CRC [14,15]. Pao and co-workers demonstrated that aberrant expression of γ-catenin resulted in phenotypic changes characteristic of transformation accompanied by genomic instability and the acquisition of properties that could contribute to colorectal tumor spread and progression [16].

MicroRNAs (miRNAs) are a family of highly conserved small (20–22 nt) noncoding RNA molecules. They bind to complementary mRNA sequences and act as regulators of gene expression, resulting in post-translational repression or mRNA degradation. Under physiological conditions, miRNAs regulate a range of biological functions, and their feedback mechanisms control key biological processes including cell proliferation, differentiation, survival, and apoptosis [17].

Accumulating evidence indicates that miRNAs contribute to colorectal tumorigenesis and metastasis, targeting the expression of key genes involved in the establishment of junctional complexes [18,19,20]. In our recent study, we reported that miR-195-5p regulates desmosome expression by targeting JUP [21]. Specifically, in vitro we demonstrated that gain of miR-195-5p significantly decreased JUP expression at the mRNA and protein levels. The downregulation of JUP by miR-195-5p, indirectly, led to overexpression of desmosome cadherins (desmoglein 2 and desmocollin 2). The intracellular increase in miR-195-5p also modulated the expression of NLK, LEF1, and cyclin D1, key effectors of the Wnt pathway. Furthermore, gain of miR-195-5p strongly reduced the migration ability of CRC cell lines [21]. All these findings suggest that JUP may be involved in the development and dissemination of colorectal cancer.

To fully elucidate the mechanisms driven by miR-195-5p in the regulation of desmosome function in CRC, we aimed to evaluate miR-195-5p and JUP expression in specimens from CRC patients collected at our institute in this study. Moreover, we investigated the effect of miR-195-5p on CRC evolution in terms of cell proliferation, viability, and invasion. Finally, we functionally characterized the in vivo effect of miR-195-5p in CRC progression.

## 2. Results

### 2.1. miR-195-5p Expression in CRC Patients

In order to assess whether miR-195-5p levels were deregulated in CRC, we analyzed miR-195-5p expression in our cohort of 30 CRC patients who underwent surgical resection. From each patient, RNA of the tumoral section and normal adjacent sample were collected. The data obtained by qPCR are reported in Figure 1 and demonstrated the critical downregulation of miR-195-5p in tumor samples compared to normal tissue (*p* < 0.0001).

### 2.2. JUP Expression in CRC Patients

JUP expression was evaluated by performing real-time PCR on RNA extracted from our cohort of CRC patients. qPCR results revealed that JUP mRNA expression levels were significantly upregulated in tumor compared to adjacent normal colon tissues (*p* < 0.05; Figure 2).

To further demonstrate our findings, we evaluated JUP protein expression in tissue specimens from the same cohort of CRC patients used to assess mRNA expression. IHC analysis highlighted that JUP expression was higher in CRC tissues compared to the corresponding peritumoral tissues (Figure 3A). Based on JUP signal intensity, we defined a immunoreactivity score ranging from 0 to 3 (absent to strong). The score reported in Figure 3B highlighted that JUP signal intensity was aberrantly increased in CRC patients.

### 2.3. miR-195-5p Reduces Cell Proliferation and Viability

To further evaluate the effect of JUP modulation by miR-195-5p on cancer progression, BrdU and CellTiter-Blue assays were performed to investigate Caco2 and LoVo proliferation and viability after miR-195-5p mimic transfection at 30 nM and 50 nM concentrations. Our results highlighted that the gain of miR-195-5p significantly inhibited cell proliferation (*p* < 0.0001; Figure 4A) and viability (*p* < 0.0001; Figure 4B) compared to mock control.

### 2.4. miR-195-5p Inhibits Cell Invasion

Further functional assays were performed to study the involvement of miR-195-5p in cell–matrix and cell–cell interactions and the invasive activity of Caco2 and LoVo cell lines. The increase in miR-195-5p significantly weakened the invasion rates of cells as compared with nontransfected cells in Caco2 (*p* < 0.0001; Figure 5A,B) and LoVo (*p* < 0.0001; Figure 5C,D) cells. These data suggest that the miR-195-5p mimic reduced the invasion ability of colon cancer cells in vitro.

### 2.5. Effect of miR-195-5p Mimic Treatment on a Mouse Model of AOM/DSS-Induced CRC 

In order to investigate the therapeutic effect of miR-195-5p in vivo, we used an Azoxymethane (AOM)/Dextran sodium sulfate (DSS)-induced CRC mouse model that mimics human CRC (Figure 6A). Starting on day 45, miR-195-5p administration significantly offset the weight loss in the treated group compared to the vehicle group (*p* < 0.05; Figure 6B).

For each mouse, macroscopic tumors were counted, and the number of tumors in the total colon were calculated for the control and treated groups. As shown in Figure 6C, after intraperitoneal administration of the miR-195-5p mimic, the number of colonic neoplasms significantly decreased in the treated group compared to the vehicle group (*p* < 0.0001). In addition, comparing tumor localization, a significant reduction in tumor numbers was observed in the medial and distal portions of treated mice compared to the vehicle group (*p* < 0.001; *p* < 0.0001; Figure 6D). Moreover, regarding the size of tumors (<2 mm, >2 mm; Figure 6D), the administration of miR-195-5p mimic in the treated group significantly reduced the number of neoformations measuring less than 2 mm (*p* < 0.001) and the rate of lesions measuring greater than 2 mm (*p* > 0.05), although the latter did not reach significance.

After intraperitoneal administration of the miR-195-5p mimic, increased expression of miR-195-5p was observed in the analyzed colon portions (medial, and distal) of treated mice compared to vehicle mice (*p* < 0.0001; Figure 6E). According to our in vitro experimental results, JUP expression in colon segments was remarkably reduced in miR-195-5p-treated mice (*p* < 0.0001; Figure 6E) compared to the control group.

To further investigate the effect of miR-195-5p in counteracting AOM/DSS-induced tumorigenesis, we evaluated miR-195-5p expression levels in the medial and distal colons of vehicle and treated mice.

Histopathological changes were confirmed in hematoxylin and eosin (H & E)-stained specimens. In stained sections, flat, nodular, polypoid, or caterpillar-like colon tumors were observed in the medial and distal colon of all AOM/DSS mice. Tumor infiltrations into the submucosal layer were also observed in AOM/DSS mice. Colon tumors of AOM/DSS mice presented noticeable pathological changes and had the typical appearance of adenocarcinoma, including round or ovoid nuclei with remarkable nucleoli, loss of nuclear polarity, inflammatory cell infiltration, and depletion of goblet cells. Treatment with miR-195-5p decreased the number of tumors and inflammatory infiltrations observed in the colonic mucosae of mice with AOM/DSS-induced colon tumorigenesis (Figure 6F). 

These data highlighted a clear improvement in the pathological colon changes in AOM/DSS treated mice and shed light on the protective effect of miR-195-5p against CRC growth and progression.

## 3. Discussion

Desmosomes are specialized anchoring junctions that in association with intermediate filaments tether cells together, provide tissues with the ability to resist mechanical forces, and stabilize their architecture [22]. In addition to their role in clustering adhesive complexes, desmosomal components exhibit signaling activities in the cytosol and nucleus, suggesting their dynamic involvement in cell and tissue function [23,24]. The complex expression pattern of desmosome proteins in epithelia and their influence on variable downstream signaling events suggest a specific contribution of individual desmosomal components to the progression of several diseases [25,26,27]. 

JUP, another vertebrate catenin of the Armadillo protein family, is a key desmosome member that was found to be overexpressed in several malignancies [28,29,30,31,32,33,34,35] and resulted in uncontrolled cell growth and foci formation due to BCL-2 induction, a prototypic member of the family of anti-apoptosis regulatory proteins [36]. Moreover, JUP strongly activates c-Myc and cyclin D1 expression [37,38] and supports cancer progression by mediating survival through the inhibition of apoptosis and promotion of cellular proliferation [15,39,40]. JUP downregulation or knockdown has been shown to suppress cell aggregation and tumor metastasis formation [35,41].

Given the emerging role of microRNAs in cancer progression, some researchers have demonstrated that miR-103 has strong tumor-promoting effects and enhances in vitro cell proliferation and migration via ZO-1 modulation [42]. Algaber and co-workers found that inhibition of FHL2 expression using a miR-340-5p mimic increased E-cadherin expression and reduced colon cancer cell migration and invasion [43]. In our recent work, we characterized the in vitro role of miR-195-5p in the regulation of desmosome junctions via JUP [21]. In detail, we found that after increasing the intracellular levels of miR-195-5p in human colonic epithelial cells, JUP expression was significantly decreased, whereas the expression of the colonic cadherins, DSG2 and DSC2, was indirectly modulated. In addition, we revealed that gain of miR-195-5p clearly inhibits the migration rate of CRC cell lines, suggesting that the ameliorative effects provided by miR-195-5p enhanced the structure and function of cell junctions. Furthermore, our results revealed that the JUP downregulation by miR-195-5p may influence NLK, LEF1, and cyclin D1 protein levels, highlighting the potential effects of miR-195-5p in Wnt pathway activation and CRC development [21].

Here, we demonstrated that miR-195-5p was downregulated in colon cancer tissues, while JUP was overexpressed in CRC tissues compared to the normal counterpart. IHC analysis underlined that JUP protein expression was higher in the entire epithelial layer of cancer tissue compared to adjacent normal samples, suggesting that JUP could be a good candidate for the treatment of CRC. In addition, we performed a broader investigation of the role of miR-195-5p in the regulation of desmosome function and CRC progression, finding that gain of miR-195-5p strongly inhibited in vitro cell proliferation, viability, and invasion. 

Finally, we verified the in vivo effect of miR-195-5p in an AOM/DSS-induced CRC mouse model. We observed that the in vivo gain of function of miR-195-5p enhanced miR-195-5p expression in the medial and distal colons of treated mice. Higher miR-195-5p levels, in turn, significantly reduced JUP expression. miR-195-5p-treated mice showed fewer tumors in the entire colon. In addition, fewer tumors were observed in the medial and distal portions of treated mice compared to the control group. miR-195-5p administration was also able to significantly reduce the number of neoformations measuring less than 2 mm and lowered the rate of lesions greater that 2 mm. These data strongly corroborated our in vitro results and further validate the role of miR-195-5p in preventing CRC progression.

Overall, our results offer further evidence of the clinical potential of miR-195-5p, and our study also points to a novel regulatory mechanism of the desmosome complex in CRC. However, the identification of JUP as its target gene and its downregulation effects in cancer remains to be fully elucidated. Indeed, future studies could be performed to further investigate the molecular mechanisms involved in the regulation of desmosome function in CRC and to clarify the effectiveness of miR-195-5p-based therapy in other in vivo CRC models.

## 4. Materials and Methods

### 4.1. Human CRC Tissues

Sixty formalin-fixed and paraffin-embedded (FFPE) tissue specimens including tumor and adjacent normal portions from CRC patients were retrospectively retrieved at the National Institute of Gastroenterology “S. de Bellis”, Castellana Grotte, Bari, Italy. Written informed consent was obtained from all participants. The study was performed according to the principles of the Declaration of Helsinki and was approved by the local institutional ethics review board (Istituto Tumori Giovanni Paolo II, Bari, Italy). Tumor and adjacent normal colon tissue were collected from each patient. Sections stained with hematoxylin and eosin were analyzed by a pathologist to assess the adequacy of the tissues and their morphologic and/or pathological characteristics. 

### 4.2. RNA Extraction and Real-Time PCR of FFPE Tissue

Total RNA, including miRNAs, from FFPE tissue was isolated using the miRNeasy FFPE kit (Qiagen, Hilden, Germany) following the manufacturer’s instructions, including treatment with deparaffinization solution (Qiagen, Hilden, Germany). 

For miR-195-5p detection, total RNA was reverse transcribed using a TaqMan Advanced miRNA cDNA Synthesis Kit (Thermo Fisher Scientific, Waltham, MA, USA), following the manufacturer’s instructions. Real-time RT-PCR for miR-195-5p quantification was performed in 20 μL of final volume using the CFX96 System (Biorad Laboratories, Hercules, CA, USA) with TaqMan Advanced miRNA assays and TaqMan Fast Advanced Master mix (Thermo Fisher Scientific, Waltham MA, USA). miR-26a-5p and miR-186-5p were used as endogenous controls to perform normalization for human and mouse data, respectively. 

For JUP analysis, total RNA was retrotranscribed using SuperScript™ VILO™ MasterMix (Thermo Fisher Scientific, Bremen, Germany). Quantitative real-time PCR was performed on a CFX96 System (Biorad Laboratories, Hercules, CA, USA) using the SsoAdvanced Universal SYBR Green Supermix (BioRad Laboratories, Hercules, CA, USA) and the QuantiTect Primer Assay for JUP and GAPDH (Qiagen, Hilden, Germany). GAPDH gene amplification was used as reference standard to normalize the relative expression of JUP. 

The relative expression was calculated using the 2^−ΔCt^ and 2^−ΔΔCt^ formulas.

### 4.3. Immunohistochemistry (IHC)

For IHC analysis of γ-catenin, 3 µm sections were cut and mounted on Apex Bond IHC slides (Leica Biosystems, Buffalo Grove, IL, USA) and then incubated with human anti-γ-catenin polyclonal antibody (#75550S, Cell Signaling, Technology, Danvers, MA, USA, 1:200 dilution) for 30 min at room temperature. γ-Catenin staining was evaluated with an automated immunostainer (Leica Biosystems, Buffalo Grove, IL, USA). Antigen retrieval was performed with BOND Epitope Retrieval Solution 2 (Leica Biosystems, Buffalo Grove, IL, USA) using EDTA pH 9. For IHC detection, the Bond Polymer Refine Detection Kit (Leica Biosystems, Buffalo Grove, IL, USA) was used as a visualization and chromogen reagent according to the manufacturer’s protocol. When the number of stained cells was less than 5%, the samples were recorded as negative. 

γ-Catenin expression was scored as follows: 0, (no staining) negative; 1, weak expression; 2, moderate expression; and 3, strong expression. 

### 4.4. Cell Culture and In Vitro Transfection

The human colonic epithelial cell lines Caco2 and LoVo were purchased from ATCC (American Type Culture Collection, Manassas, VA, USA) and were maintained in Dulbecco’s Modified Eagle Medium (DMEM, Thermo Fisher Scientific, Waltham, MA, USA) supplemented with heat-inactivated fetal bovine serum (10% for Caco2 and 20% for LoVo) (FBS, Thermo Fisher Scientific, Waltham, MA, USA), 1% streptomycin/penicillin (Thermo Fisher Scientific, Waltham, MA, USA), 10 mM HEPES (Sigma-Aldrich, St. Louis, MO, USA) and 1 mM sodium pyruvate (Sigma-Aldrich, St. Louis, MO, USA). For viability, proliferation, migration, and invasion assays, cells were initially seeded in 6-well plates and transfected with miR-195-3p mimic (Life Technologies, Carlsbad, CA, USA) at a concentration of 30 nM and 50 nM using TransIT-TKO Transfection Reagent (Mirus Bio LLC, Madison, WI, USA) according to the manufacturer’s instructions. This compound is a liposomal-based transfection reagent that enables the formation of positively charged lipid aggregates that merge smoothly with the phospholipid bilayer of the host cell to allow the entry of the foreign genetic materials with minimal resistance. miR-195-5p and lipid reagent were diluted using unsupplemented basal medium and kept at room temperature for 30 min for complexing before being administered to cells. Each transfection experiment was normalized with mock control, corresponding to cells transfected with empty liposome. 

### 4.5. Proliferation and Viability Assays

Proliferation was measured using the cell proliferation ELISA, BrdU (colorimetric) assays provided by Roche Applied Biosciences (Laval, QC, Canada) following the manufacturer’s instructions. Briefly, cells were harvested 48 h post-transfection, counted, and reseeded in 96-well plates (Corning, Corning, NY, USA). Then, cells were incubated for 3 h in the presence of BrdU before fixing and labeling with anti-BrdU antibody (90 min). The relative absorbance was detected using a microplate reader (SPECTROstar Omega, BMG Labtech, Ortenberg, Germany).

For viability assays, after miR-195-5p mimic transfection, Caco2 and LoVo cells were harvested and reseeded in 96-well plates. Cells viability was assessed using the CellTiter-Blue Cell Viability Assay (Promega, Madison, WI, USA) according to the manufacturer’s recommendations. Cells were incubated with Cell Titer Blue Reagent at 37 °C for one hour, and the relative fluorescence (560 Excitation/590 Emission) was measured with a FLUOstar Omega microplate spectrophotometer.

### 4.6. Invasion Assays

For the transwell assay, the polycarbonate membranes of the upper surface of the 6.5 mm Transwell^®^ inserts (8 µm pore size; Corning, Corning, NY, USA) were precoated with Matrigel.

After transfection, Caco2 and LoVo cells were suspended in 200 µL of serum-free medium and reseeded into the upper chamber of each insert. Then, 500 μL of medium supplemented with 10% FBS was added to the lower chamber. After 48 h, cells that invaded through the membrane were fixed using 4% paraformaldehyde for 15 min and stained with 0.1% crystal violet (Sigma-Aldrich, St. Louis, MO, USA) for 15 min. Cells on the top surface of the insert were removed with gentle wiping using a cotton swab. The fixed cells in the bottom surface of the inserts were counted in randomly selected fields using a light microscope. Each experiment was independently repeated at least three times.

### 4.7. Animal Experiments

The animal experiments were performed under the guidance of ethical standards and in accordance with national and international guidelines. The study was approved by the authors’ institutional review board (Organization for Animal Wellbeing [OPBA]). 

All animal experiments were conducted according to Directive 86/609 EEC required by Italian D.L. n. 26/2014 and approved by the Committee on the Ethics of Animal Experiments of the Ministero della Salute—Direzione Generale Sanità Animale (Authorization n. 616/2022-PR) and the official RBM veterinarian. In order to safeguard the animals’ welfare, in cases of severe clinical conditions, the animals were sacrificed to avoid further suffering. The cages for animals permitted exercise and normal social behaviors and were well ventilated, softly lit, and subject to a light dark cycle. The relative humidity was kept at 45 to 65%, and the temperature was constantly maintained at 20–25 °C. Each standard cage was provided with food, water, dry bedding, and nesting material in order to keep their cage clean with minimal disturbance and stress when cleaning.

### 4.8. AOM/DSS-Induced CRC Model and Treatment

Male C57BL/6 mice (n = 28; 7 weeks old; 19–25 g) were divided into two groups: vehicle group (n = 14) and treated group (n = 14). For the AOM/DSS-induced CRC mouse model [44,45], 12 mg/Kg of azoxymethane (AOM; Sigma-Aldrich, St. Louis, MO, USA) dissolved in PBS was administered to each animal by oral gavage. After 5 days, drinking water containing 2% (*w*/*v*) dextran sodium sulfate (DSS; 36–50 KDa; MP Biomedical, Santa Ana, CA, USA) was added for 5 days, followed by distilled water for 15 days. For the next two cycles, mice were exposed to 1% (*w*/*v*) DSS. 

For in vivo transfection, miR-195-5p mimic (0.5 nmol/mice), previously complexed with Invivofectamine 3.0 Reagent (Thermo Fisher Scientific, Waltham, MA, USA), was administered intraperitoneally to the treated group. For the vehicle group, the volume of miRNA mimic was replaced with an equal volume of PBS. The injections were started on day 26 and administered twice a week until sacrifice. Body weight was monitored daily after AOM administration. On day 85, mice were sacrificed, and the colon portions (proximal, medial and distal) were collected for molecular and histopathological analysis. Total RNA and small RNA from tissues were extracted using TRIzol reagent (Invitrogen, Carlsbad, CA, USA), according to the manufacturer’s protocol.

### 4.9. Histology

The mouse colon specimens were fixed in 4% paraformaldehyde and embedded in paraffin using standard procedures. Briefly, 3 μm sections of fixed colon segments were deparaffinized, stained with hematoxylin and eosin (H & E), and examined blinded by a pathologist to evaluate the morphologic and/or pathological characteristics of the samples.

### 4.10. Statistical Analysis

Statistical analysis was performed using GraphPad Prism software version 9.0.0. Statistical significance of data resulting from different conditions was evaluated with a two-tailed Student’s *t* test. All values are expressed as the mean ± SEM of data obtained from at least three independent experiments. Differences among experimental conditions were considered statistically significant at *p* < 0.05.

## 5. Conclusions

In conclusion, we demonstrate miR-195-5p and JUP dysregulation in CRC patients. miR-195-p inhibits CRC growth and progression in vitro as well as in an experimental CRC mice model via JUP downregulation. These findings could have potential clinical relevance for future miRNA-based therapies in CRC.

## Figures and Tables

**Figure 1 ijms-25-00494-f001:**
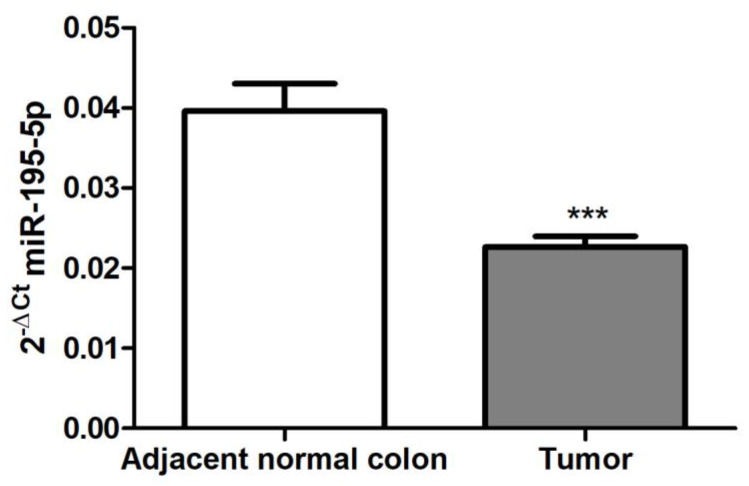
miR-195-5p expression in tumor and adjacent normal colon tissues. qPCR performed on CRC patients specimens (n = 30) that include tumor and adjacent normal colon tissues to evaluate miR-195-5p expression levels. Our data highlighted that miR-195-5p was significantly downregulated in CRC tissue compared to normal samples. *** *p* < 0.0001.

**Figure 2 ijms-25-00494-f002:**
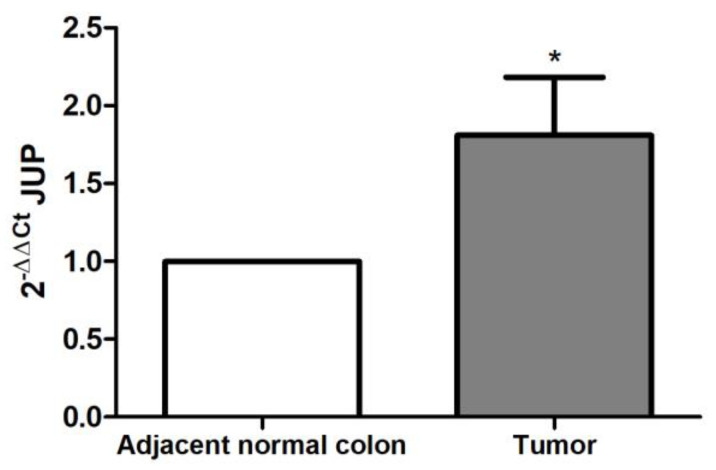
JUP mRNA expression in CRC and adjacent normal colon tissues. Real-time PCR assessment using formalin-fixed and paraffin-embedded tissue blocks from CRC patients. JUP mRNA expression levels were significantly higher in tumor compared to healthy control tissues. * *p* < 0.05.

**Figure 3 ijms-25-00494-f003:**
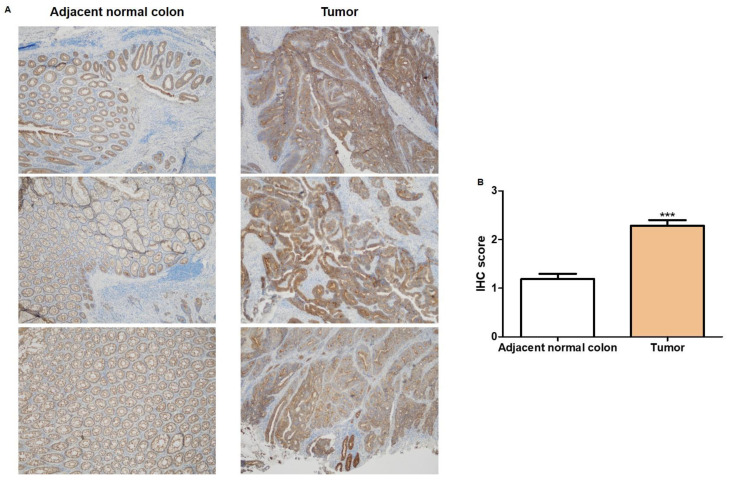
γ-Catenin protein expression in CRC patients. (**A**) Representative images derived from distinct FFPE blocks of intestinal tissues show increased immunoreactivity of γ-catenin in tumor compared to adjacent normal colon tissues. Original magnification, ×4. (**B**) IHC score representing the expression levels of γ-catenin in the intestinal epithelium ( *** *p* < 0.001).

**Figure 4 ijms-25-00494-f004:**
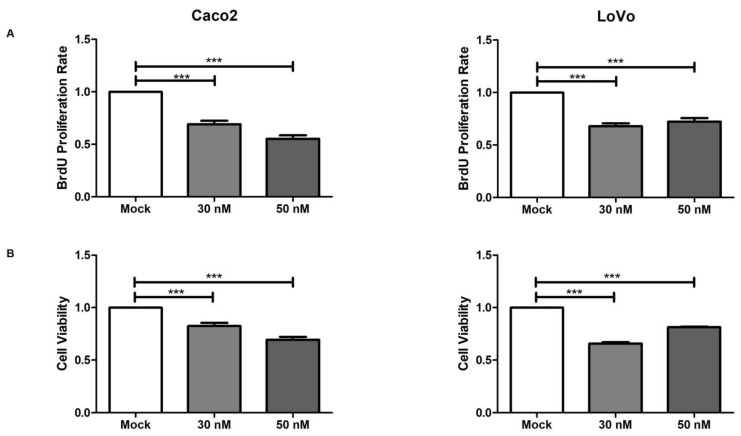
miR-195-5p reduces cell proliferation and viability. (**A**) Colorimetric immunoassay for the quantification of cell proliferation. The increase in miR-195-5p regulated the incorporation of BrdU during DNA synthesis, resulting in reduced cell growth. The proliferation rate showed a significant reduction under transfected conditions in all cell lines, with the mock-control demonstrating the maximum amount of BrdU incorporation (100%) at 3 h of incubation. (**B**) Viability of Caco2 and LoVo cells assessed using the CellTiter-Blue assay after miR-195-5p mimic transfection. Cell viability was significantly reduced in both transfected conditions compared to mock control. Data are presented as relative levels of proliferation and viability and are the mean of four independent experiments ± SEM. *** *p* < 0.0001.

**Figure 5 ijms-25-00494-f005:**
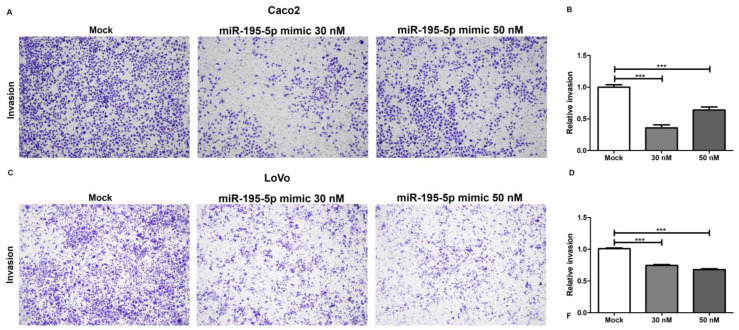
Effect of the miR-195-5p mimic on Caco2 (**A**,**B**) and LoVo cell (**C**,**D**) invasion. Under transfection conditions, cell aggressiveness was significantly reduced. For each cell line, the acquired images of invading cells and the relative invasion rate are shown. Magnification 4×.The results are presented as relative level of invasion and as the mean of at least three independent experiments ± SEM. *** *p* < 0.0001.

**Figure 6 ijms-25-00494-f006:**
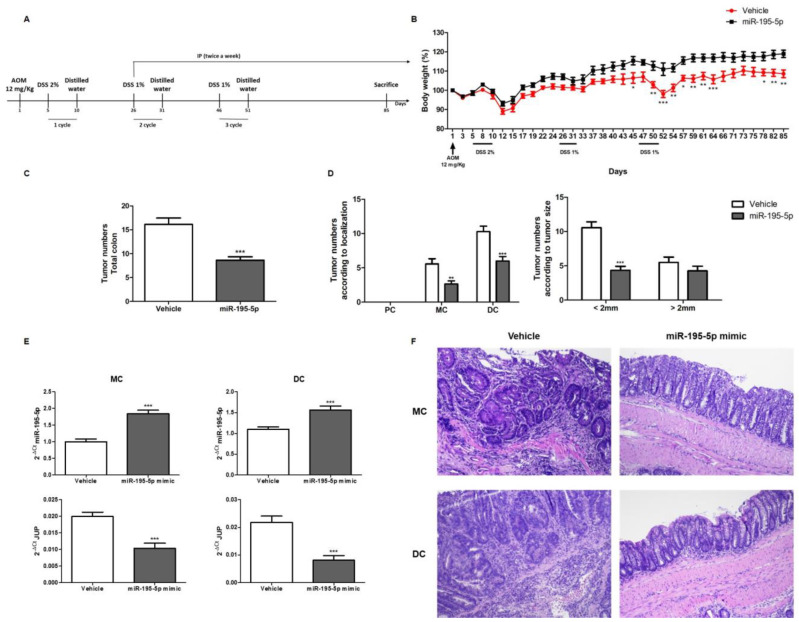
Effect of miR-195-5p mimic treatment on pathological features in the AOM/DSS-induced CRC mouse model. (**A**) Schematic diagram of AOM/DSS-induced CRC mice and miR-195-5p administration (n = 14 mice/group). (**B**) For each group, body weight was monitored daily after AOM administration for the duration of treatment. Significant differences between groups were determined using two-way ANOVA tests. (**C**) Total number of tumors in each group was assessed. Values are the mean ± SEM (n = 14 mice/group). (**D**) Total numbers of tumors according to colon localization and tumor size were determined. (**E**) miR-195-5p and JUP expression in the medial and distal colons of treated and untreated mice. (**F**) Representative images of hematoxylin and eosin staining of medial and distal colon tissue from each group obtained at 10× magnification. * *p* < 0.05, ** *p* < 0.01, *** *p* < 0.0001.

## Data Availability

Data are contained within the article.

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
