# Peer review of "Downregulation of γ-Catenin by miR-195-5p Inhibits Colon Cancer Progression, Regulating Desmosome Function"

_ijms, 2023, doi:10.3390/ijms25010494_

Round 1

Reviewer 1 Report

Comments and Suggestions for Authors

The study by Dr Piccinno presents interesting results on the role of miR-195-5p in inhibiting colorectal cancer growth and progression, throught the regulation of JUP and desmosome function.

The paper is well-written and sufficiently complete.

However, some clarifications and minor revisions are needed before publication:

-       - Authors should make uniform through the text the nomenclature of JUP or γ-catenin

-      - How the proliferation rate was calculated in the experiment of BrdU incorporation is not sufficiently clear. Please described it in the Methods chapter or in the Figure 4 legend.

-      - Some details about miR-195-5p transfection should be added.

-   How were the endogenous controls for miR detection selected? Are authors sufficiently sure that only one miR as reference is enough to produce accurate results?

Author Response

The study by Dr Piccinno presents interesting results on the role of miR-195-5p in inhibiting colorectal cancer growth and progression, throught the regulation of JUP and desmosome function.

The paper is well-written and sufficiently complete.

However, some clarifications and minor revisions are needed before publication:

We thank the referee for the critical and helpful evaluation and for the opportunity given to us to revise it.

-       Authors should make uniform through the text the nomenclature of JUP or γ-catenin

We used the nomenclature of JUP or γ-catenin to distinguish between the gene (JUP) and the relative protein product (γ-catenin).

-      How the proliferation rate was calculated in the experiment of BrdU incorporation is not sufficiently clear. Please described it in the Methods chapter or in the Figure 4 legend.

We thank the reviewer for the suggestion, and we added this information in Figure 4 legend.

-      Some details about miR-195-5p transfection should be added.

As suggested by referee, we have added it in “Materials and Methods” section.

-  How were the endogenous controls for miR detection selected? Are authors sufficiently sure that only one miR as reference is enough to produce accurate results?

In this work, the TaqMan Advanced miRNA Assays was used for real time PCR measurements and this kit do not detect snRNAs or snoRNAs. According to manufacturer’s protocol, from a list of endogenous controls provided by Thermo Fisher Scientific, we choose miRNA-26a-5p as reference since it was one of the most stable miRNAs in CRC (Chang, KH et al. MicroRNA expression profiling to identify and validate reference genes for relative quantification in colorectal cancer. BMC Cancer 10, 173 (2010) (https://doi.org/10.1186/1471-2407-10-17).

Reviewer 2 Report

Comments and Suggestions for Authors

1.       How is the selected microRNA involved in the cancer pathogenesis and mechanism?

2.       Gain of function and loss of function and Lucifer method need to find the role of the microRNAs.

3.       The author needs to normalize the sample and then analyze the data based on nonparametric or parametric analysis.

4.       What is the take-home message of this manuscript?

5.       Describe the limitations and suggestions for future study. Moreover, use these papers:

https://doi.org/10.3390/jpm12030456

https://doi.org/10.1016/j.envres.2023.116980

https://doi.org/10.3390/biology12111426

https://doi.org/10.1016/j.envres.2023.117168

6.       What is your novelty?

7.       The author has to explain the practical application of this study.

8.       AOM/DSS-Induced CRC Model and Treatment: need references.

9.       The animal working is not complete. The housing feature needs to explain.

Author Response

We thank the referee for the critical and helpful evaluation and for the opportunity given to us to revise it.

  1. How is the selected microRNA involved in the cancer pathogenesis and mechanism?

In our previous works (doi: 10.3390/biomedicines10040919; https://doi.org/10.3390/ijms23105840) we have in vitro and in vivo characterized the role of miR-195-5p in the regulation of Tight Junction. Hence, in this study we aimed to investigate the effect of miR-195-5p in the modulation of other key cell junctions.

miR-195–5p had been found to correlate with almost all types of human cancers, which included respiratory system, digestive system, urinary system, nervous system, motor system, reproductive system and endocrine system (https://doi.org/10.1016/j.biopha.2022.112885). miR-195–5p targeted many oncogenes, and was thus implicated in various tumorigenic processes such as migration, invasiveness, proliferation and chemoresistance. (https://doi.org/10.1042/BSR20191850; https://doi.org/10.1159/000494602; https://doi.org/10.1186/s12943-018-0933-7; https://doi.org/10.2147/PGPM.S302755). In addition, low miR-195–5p expression was frequently observed in various tumors and was shown to be closely correlated with advanced clinical stage, metastasis, invasiveness, chemoresistance and unfavorable prognosis (https://doi.org/10.1186/s12935-020-01702-0; https://doi.org/10.1177/1533033819887962; https://doi.org/10.1002/jcp.29016; https://doi.org/10.1186/s13046-021-02027-0). Therefore, miR-195–5p has the potential to be biomarkers for cancer diagnoses, predicting treatment outcomes and patient prognosis.

Moreover, in our recent study, we reported that miR-195-5p and JUP was able to regulate desmosome expression, targeting JUP [REF 21]. Specifically, in vitro we demonstrated that the gain of miR-195-5p was able to significantly decrease the JUP expression at mRNA and protein levels. This regulation, in turn, determined an indirect modulation of desmosome cadherins (Desmoglein 2 and Desmocollin 2). miR-195-5p gain of function was also able to modulate the expression of key components of the Wnt pathway such as NLK, LEF1 and Cyclin D1. Furthermore, the increase of the intracellular levels of miR-195-5p significantly inhibited the migration ability of CRC cell lines. All these findings demonstrate that JUP may be involved in the development and dissemination of colorectal cancer. To fully elucidate the mechanisms controlled by miR-195-5p in the regulation of desmosomes function in CRC, in this study we aimed to evaluate the miR-195-5p and JUP expression in specimens from CRC patients collected at our institute. Moreover, we investigated the effect of miR-195-5p on CRC evolution in terms of cell proliferation, viability and invasion. Finally, we functionally characterized the in vivo effectiveness of miR-195- 5p in CRC progression.

  1. Gain of function and loss of function and Lucifer method need to find the role of the microRNAs.

We agree with the reviewer, but this topic has been investigated in our recent study in which we demonstrated the affinity between miR-195-5p and JUP, an essential desmosome component that strictly interact with desmosomal cadherins (REF 21). We have also reported that miR-195-5p and JUP was able to regulate desmosome expression, targeting JUP (REF 21). Specifically, in vitro we demonstrated that the gain of function of miR-195-5p was able to significantly decrease the JUP expression at mRNA and protein levels. This regulation, in turn, determined an indirect modulation of desmosome cadherins (Desmoglein 2 and Desmocollin 2). miR-195-5p gain of function was also able to modulate the expression of key components of the Wnt pathway such as NLK, LEF1 and Cyclin D1. Furthermore, the increase of the intracellular levels of miR-195-5p significantly inhibited the migration ability of CRC cell lines (REF 21).

  1. The author needs to normalize the sample and then analyze the data based on nonparametric or parametric analysis.

For qPCR analysis, miR-195-5p expression were normalized using miR-26a-5p as endogenous control while GAPDH gene amplification was used as reference standard to normalize the relative expression of JUP. In addition, the data derived from proliferation, viability and invasion assays were normalized on mock control values.

Our data showed a Gaussian distribution and therefore in the statistical analysis we used a parametric test as Student’s t-test.

  1. What is the take-home message of this manuscript?

In this study, we aimed to experimentally demonstrate for the first time a miR-195-5p and JUP dysregulation in CRC patients. Furthermore, our work highlighted that miR-195-p gain of function was able to significantly inhibit CRC growth and progression in vitro as well as in an experimental CRC mice model via JUP downregulation. We believe that these findings could have a potential clinical relevance for future miR195-5p-based therapy in CRC.

  1. Describe the limitations and suggestions for future study. Moreover, use these papers:

https://doi.org/10.3390/jpm12030456

https://doi.org/10.1016/j.envres.2023.116980

https://doi.org/10.3390/biology12111426

https://doi.org/10.1016/j.envres.2023.117168

We thank the reviewer for the suggestion and we added the limitations and suggestions for future study in “Discussion” section.

Our experimental and clinical evidence points to miR-195-5p and JUP as key elements in cancer development, since their involvement in cell invasion, proliferation, and viability. Our in vitro and in vivo studies have shown that JUP inhibition by miR-195-5p has positive effects on cancer therapy, which enhances its role as a potential therapeutic target. However, the identification of JUP as its target genes and its downregulation effects in cancer remains to be fully elucidated. Indeed, future studies could be performed to further investigate the molecular mechanisms involved in the regulation of desmosome function in CRC and to clarify the effectiveness of miR-195-5p-based therapy in other in vivo CRC models.

  1. What is your novelty?

To the best of our knowledge at date our work showed for the first time a miR-195-5p and JUP dysregulation in CRC patients. In addition, no other works demonstrated the role of miR-195-5p in the regulation of cell proliferation, viability and invasion of CRC cell lines. As novelty, we observed that the in vivo gain of function of miR-195-5p significantly reduced JUP expression and miR-195-5p-treated mice showed fewer tumors in the entire colon. miR-195-5p administration was also able to significantly reduce the number of neoformations measuring less than 2 mm and lowered the rate of formations over 2 mm. These data strongly corroborated our in vitro results and further validate the role of miR-195-5p to contrast CRC progression.

  1. The author has to explain the practical application of this study.

Our work shed the light for future clinical studies. Currently, miR-195-5p based therapy is not still being experimented in clinical trials and toxicity pharmacokinetic evaluations are not still being conducted for miR-195-5p. Despite these limitations and to the lack of knowledge miRNAs in cancer, the administration and delivery of miRNA mimics and antimiRs, are being solved since various oncology clinical trials using miRNAs in screening, diagnosis, and drug testing are currently underway.

  1. AOM/DSS-Induced CRC Model and Treatment: need references.

We thank the reviewer for the suggestion and we have added the references.

  1. The animal working is not complete. The housing feature needs to explain.

According to the referee suggestion, we added information about housing features in “Materials and Methods” section.

Round 2

Reviewer 2 Report

Comments and Suggestions for Authors

The author responds to the comments.